# Dry Anaerobic Digestion of Chicken Manure: A Review

**Yevhenii Shapovalov** [1,*] , **Sergey Zhadan** [2] , **Günther Bochmann** [3] , **Anatoly Salyuk** [4] and **Volodymyr Nykyforov** [5]

1   Department of Knowledge systems creation, National Center of «Junior Academy of Science of Ukraine», 04119 Kyiv, Ukraine

2   Individual Entrepreneur "Dyba", 03035 Kyiv, Ukraine; zhadan@nuft.edu.ua

3   Institute for Environmental Biotechnology, University of Natural Resources and Life Sciences, 1180 Vienna, Austria; guenther.bochmann@boku.ac.at

4   Educational and Scientific Institute of Food Technology, National University of Food Technology, 01601 Kyiv, Ukraine; salyuk@nuft.edu.ua

5   Department "Biotechnologies and Bioengineering", Kremenchuk Mykhailo Ostrohradskyi National University, 39600 Kremenchuk, Ukraine; v-nik@kdu.edu.ua

*   Correspondence: sjb@man.gov.ua; Tel.: +380-689011575

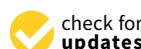

**Featured Application:  The paper could be useful for further researches in the field of dry anaerobic digestion of high nitrogen-containing waste and it can be used for optimization of an industrial setting.**

**Abstract:** Providing anaerobic digestion is a prospective technology for utilizing organic waste, however, for waste with a high content of nitrogen such as manure, dilution is necessary to decrease the ammonia inhibition effect which leads to the production of a huge effluent amount which is difficult to use. Dry anaerobic digestion has some advantages such as reduced reactor volume, higher volumetric methane yield, lower energy consumption for heating, less wastewater production, and lower logistic costs for fertilizers. These factors generate interest in using it for treatment of even high-nitrogen substrates. The purpose of this work was to analyze different dry anaerobic digestion technologies, the features of dry anaerobic digestion, laboratory studies on chicken manure dry anaerobic digestion, and methods of reducing inhibitors' effects. Nowadays, there are no dry anaerobic industrial plants working on chicken manure. However, studies on dry anaerobic digestion of chicken manure have proven the possibility of methane production under fermentation of chicken manure with high total solids content, but the process has been described as being unstable. Co-fermentation, ammonium/ammonia removal, and adaptation of the microbial consortium have been used to decrease the effect of ammonia inhibition. A prospective way for ammonia concentration control is absorption using a non-volatile sorbent located in the reactor. It decreases ammonia content during wet anaerobic digestion by 33% and it is characterized by having a positive economic effect. Therefore, dry anaerobic fermentation of chicken manure is possible, but there is still no efficient way to provide it. The results of this article should be helpful in the selection of anaerobic digestion technology for treating chicken manure.

**Keywords:** anaerobic digestion; water consumption; inhibition; ammonia; ammonium; co-fermentation; ammonia removal

## 1. Introduction

Poultry farming is one of the most advanced branches of the agroindustry complex. However, egg and chicken meat productions generate waste that needs to be utilizing. One of the most economically attractive methods for utilizing this waste is anaerobic digestion (AD). This approach produces biogas, which can be used for heating or electricity production.

Manure has a high content of organic nitrogen that is converted to ammonia (Table 1). Due to this fact, it is necessary to dilute manure with water. The amount of wastewater is directly proportional to the amount of water required for the predilution. The effluent can be used as organo-mineral fertilizer. However, in practice, using the effluent that is produced is complicated, due to the high capacity of modern poultry farms. Liquid fertilizer is difficult to store, transport, and to sell. Sometimes, it is necessary to keep effluent for periods during which it cannot be used as fertilizer. It may need to be preserved for up to a year [1,2]. Fertilizers placed in lagoons can pollute the environment [3,4]. Compared to traditional chemical fertilizers, transportation of an effluent is very expensive due to its high-water content. The distribution of a significant amount of liquid fertilizer among several small farms can be a problem [3]. Processing and reuse of liquid effluent from large biogas plants in an accessible and environmentally safe way, still remains to be an engineering problem [5].

**Table 1.** Comparison of dry anaerobic digestion (AD) technologies [6].

| Parameters | Mean Value | | A Range of Values |
|---|---|---|---|
| | g/kg | TS, % | g/kg |
| Water | 657 | – | 369–770 |
| C | 289 | 84.26 | 224–328 |
| Total N | 46 | 13.41 | 18.2–72 |
| Organic N | 38 | 11.08 | – |
| Ammonium | 14.4 | 4.20 | 0.21–29.9 |
| $NO_3$-N | 0.4 | 0.12 | 0.03–1.5 |
| Total P | 20.7 | 6.03 | 13.5–34 |
| K | 20.9 | 6.09 | 12.5–32.5 |
| Cl | 24.5 | 7.14 | 6–60 |
| Ca | 38.9 | 11.34 | 36.2–59.6 |
| Mg | 4.7 | 1.37 | 1.8–6.6 |
| Na | 4.2 | 1.22 | 2–7.4 |
| Mn | 0.3 | 0.09 | 0.26–0.38 |
| Fe | 0.32 | 0.009 | 0.08–0.56 |
| Cu | 0.53 | 0.02 | 0.04–0.07 |
| Zn | 0.35 | 0.10 | 0.29–0.39 |
| As | 0.03 | 0.01 | – |

Thus, the development of technologies that reduce water consumption and effluent formation seems to be relevant. The amount of effluent can be reduced by recirculation of the liquid phase or through a process with low moisture content. However, both approaches are known to have problems with inhibitors accumulation. This research is devoted to the analysis of different dry AD technologies, features of dry AD, laboratory studies of chicken manure dry AD, and methods of reducing inhibitors' effects.

## 2. General Aspects of Dry Anaerobic Digestion

### 2.1. Dry and Wet Anaerobic Digestion

AD can be performed as dry AD and wet AD, i.e., at different moisture content values, because under a specific content of total solids (TS), the substrate loses its fluidity. There is no generally accepted distribution limit of moisture content for dry and wet fermentation. Some authors

have defined this limit to be equal to 15% [7,8]. According to Li et al., AD was divided into wet, semi-dry, and dry with TS content ups to 10%, 10–15%, and more than 15%, respectively [9].

Similarly, there is an opinion among several authors that TS concentration during AD can be low (up to 15%), medium (15–20%), and high (20–40%) [10,11]. Lissens et al. pointed out that wet AD occurred at a TS concentration between 10 and 15%, and dry AD occurred at a TS concentration between 25 and 40% [12]. However, there was another opinion that dry AD reactors (ADR) were intended for digestion of substrates with a TS concentration between 20 and 40% [13–15]. According to Salyuk et al., the boundary between dry and wet AD for chicken manure was 16% of TS [16,17].

## 2.2. Advantages of the Dry Anaerobic Digestion

Dry AD technologies require from four to ten times [18,19] less water for dilution than wet AD technologies. Thus, advantages for dry ADR include reduced reactor volume, higher volumetric methane yields, lower energy consumption for heating, a positive energy balance, less wastewater, and, consequently, lower logistics costs for fertilizers [8,9,13,20–22], and only very dry substrates with a TS content of more than 50% require dilution [14]. One of the specific advantages of dry AD is the lack of foam [22]. Using dry ADR also guarantees decontamination of the effluent. Similar to wet AD, about 30% of energy is consumed for bioreactors heating [18].

Reduced costs for dry AD are a consequence of lower reactor volumes and savings on sorting equipment, as dry AD technologies are more stable and resistant to stones, glass, metals, plastics, and wood [13,20,22]. The process of fermentation requires removal of only very coarse particles with a size of more than 5 cm, which reduces the costs of sorting equipment [13].

The main problem of dry AD systems of municipal solid waste, agricultural waste, and food waste are mixing and transportation, but not biochemical constraints. Equipment for shipping of solids, as a rule, is more expensive than that for liquids. Shipping is carried out using conveyor belts, augers, and powerful pumps, especially designed for high-viscosity flows [22].

Perfect contact of biomass and a substrate may not be achieved due to a lack of mixing [23]. According to Abbassi-Guendouz et al., mixing was complicated when the TS content was more than 30% [24]. In addition, during dry anaerobic digestion, percolation was observed, and it was possible to intensify the process by adding straw and wood chips to save water, which could have led to an increase of the specific methane yield by 6% and 11% [25], respectively.

There are different types of processes in various parts of the reactor due to such mixing. Local inhibition and incomplete fermentation of the substrate can take place [24,26–28]. Additionally, because of mixing problems during dry AD, the process has a low rate of reaction, which may be caused by the delay of the dense substrate hydrolysis [2,24,27]. Kothari et al. [7] pointed out that inhibition of the process occurred at a TS concentration level of more than 40%, due to problems with contact between phases of the substrate and microorganisms. A high TS concentration and dense substrates can lead to obstacles with substrate heating [22].

Dry ADR requires more inoculum [8]. The dry AD process significantly depends on the activity of the inoculum, which, ideally, should be taken from biogas digesters used for fermentation of such substrates [29].

## 2.3. Inhibition of AD by Ammonia

The main problem of providing dry AD of chicken manure is inhibition due to the high content of nitrogen- [30] and sulfur-containing [31] compounds formatted in the process of hydrolysis. However, it has been determined that chicken manure digestion is primarily limited by the content of nitrogen-containing compounds [32–34], which can be present in a substrate in the forms of ammonium and non-dissociated ammonia [35]. Under dry AD, inhibition of anaerobic digestion by ammonia is much more intensive, which limits its use. However, the advantages of dry AD generate interest in its use with high-nitrogen substrates. Ammonia also causes engine corrosion and environmental pollution [36,37]. To decrease the effect of ammonia inhibition, it is necessary to provide an anaerobic

digestion process with high hydraulic retention time (HRT) [38] and, consequently, low organic load rate (OLR) [39].

Ammonia has a strong inhibitory effect on methanogens, especially on acetate-utilizing methanogens [40]. The most sensitive to increasing ammonia concentration are representatives of Methanomicrobiales, particularly the Methanoculleus sp. [41]. Methanosaetaceae are also more sensitive to inhibition by ammonia than representatives of Methasorsarcinaceae [42].

According to Qiao et al., chicken manure consists of 35.16% C, 4.83% H, 30.12% O, 5.44% N, and 0.84% S and its gross formula is $CH_{1.648}O_{0.642}N_{0.133}S_{0.009}$. Therefore, decomposition of 1 g of volatile solids (VS) of chicken manure delivers 0.087 g of $NH_3$ (0.072 N-$NH_4^+$) [43]. Therefore, the production of $NH_3$ is estimated to be from 39 to 46 g/kg (32–38 N-$NH_4^+$ g/kg) in the case of 25% TS and 60–70% VS of chicken manure. According to McCarty [44], the anaerobic digestion is inhibited at any pH when ammonium content is more than 3 g/L. However, even a concentration of ammonium from 1.5 to 2.5 g/L causes inhibition [45]. It has been proven that the process is unstable when $NH_3$-N content is more than 1 g/L [41]. Figure 1 shows potential ammonia content in case of full decomposition at different TS contents.

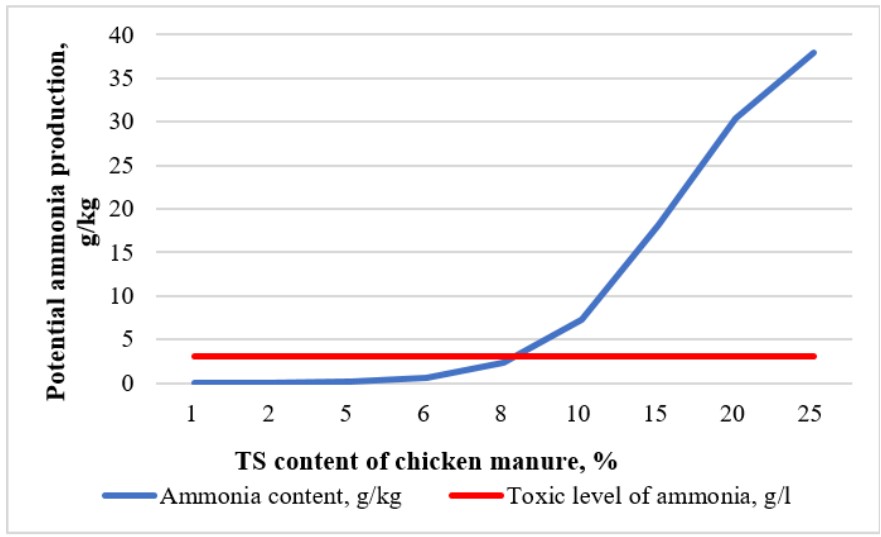

**Figure 1.** Potential ammonia concentration in case of full decomposition at different total solids (TS) contents.

As shown in Figure 2, potential ammonium content in the case of full decomposition at conditions of dry anaerobic digestion (TS content more than 16 %) is much higher than the safe level.

Mechanism of Inhibition by Ammonia

The most toxic substance for anaerobic bacteria is non-dissociated ammonia. It is diffused inside a cell and ionized to $NH_4^+$. This process leads to pH imbalance inside and outside a bacteria cell, which provides incensing of cell energy usage, decreasing of fermentative activities, violation of the transport of substances, proton imbalance, or potassium deficiency [43,46,47]. Inhibition of anaerobic metabolism can also be due to the accumulation of volatile fatty acids [48]. Huang et al. found that an increase in ammonium concentration and increased temperature led to a decrease in the proteolytic activity of the anaerobic consortium [49].

It has been proven that increasing both temperature [50,51] and pH [30] has led to an increased inhibition effect by ammonia. Figure 2 shows the proportion of free ammonia depending on pH and ambient temperature.

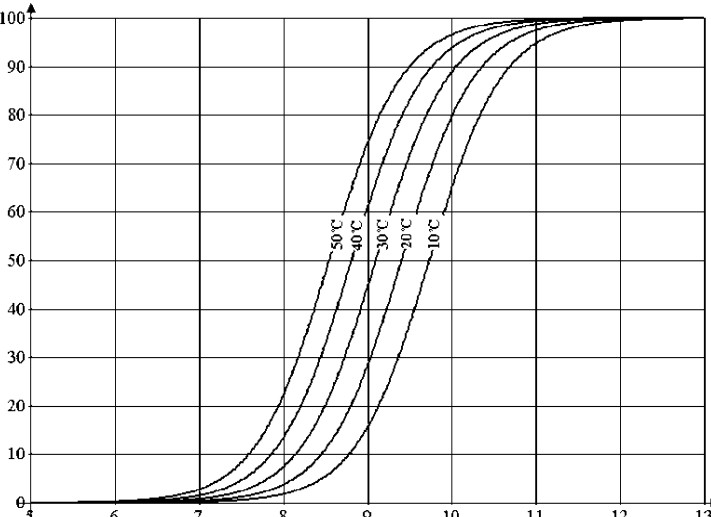

**Figure 2.** The proportion of free ammonia depending on pH and ambient temperature.

### 2.4. Influence of Water Activity on Dry AD

There may not be enough free moisture to conduct dry AD with a high TS concentration in a substrate [29]. The production of biogas is reduced in proportion to a TS concentration increase from 10 to 25%. Anaerobic digestion is unstable at a TS concentration equal to 30%, and dry AD with a TS concentration of 35% is completely suppressed [52]. Even under thermophilic conditions (where moisture is extra bioavailable), the amount of biologically available water is insufficient [53]. The same effect on $CH_4$ production of water availability is observing for anaerobic processes in soils [54].

Water activity is the ratio of the vapor pressure of the water in the material to that of pure water at a given temperature [55]. According to Corry, this is the availability of free water in the system [56]. It means that lower water activity of the substrate leads to decreasing available moisture for microorganisms [55]. Water distribution, which depends on both the water content and the water interactions with the medium compounds, determines water bioavailability and kinetic rates [57]. Biological activity can be used to estimate the possibility of using anaerobic digestion for treating waste [57]. The presence of biologically active water can affect population growth by increasing the entropy of the cell systems [58].

It has been proven that anaerobic digestion of substrates with a TS concentration of 20% lacked biologically active moisture due to low water activity for the formation of biogas by microorganisms at 20, 37, and 55 °C [29].

The activity of most microorganisms is inhibited when the water activity value is less than 0.6. Bacteria, unlike yeast and micromycetes, are less osmotolerant and their optimum growth is reached with a value of water activity in the range of 0.99–0.995 [55]. Bacteria growth is possible in the range of water activity of 0.85–1 [59]. According to Rockland and Beuchal [60], the value of water activity for high-speed hydrolysis should be more than 0.91.

An increase of TS concentration from 5 to 19% leads to a sharp decline in water activity from 0.956 to 0.93, and below the last value, water activity decreases more smoothly. Thus, the value of water activity at a TS concentration of 25% is 0.928, and at a TS concentration of 35%, it is 0.925 [61].

Water activity is affected by temperature. An increase in temperature leads to increased water activity. Thus, growth of microorganisms under thermophilic conditions is up to 60% more intense as compared with the mesophilic ones, in particular, due to higher water activity. There is more biologically active moisture at 55 °C than at 35 °C [29]. However, it is often impossible to use high temperatures, due to inhibition by ammonium, as described in Figure 3.

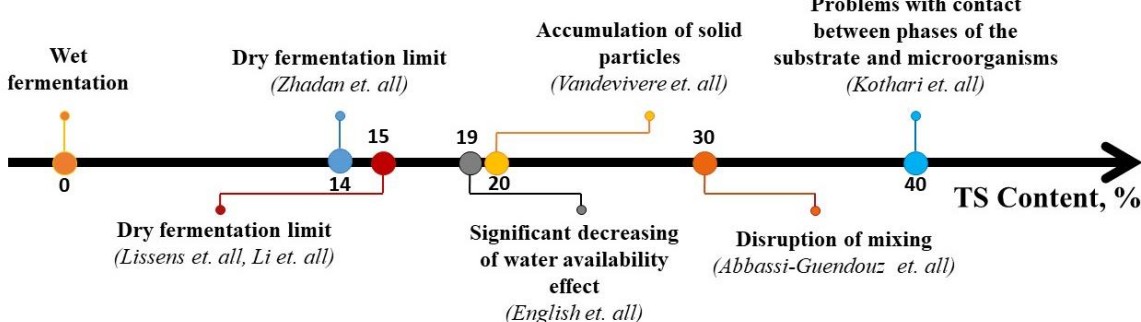

**Figure 3.** Influence of TS increase on the methane fermentation of chicken manure.

An increase in water activity under dry AD is possible by spraying water into the upper part of the reactor [29]. Thus, Rajagopal et al. [62] studied top-down and down-top modes of fluid motion under dry AD and found that the top-down mode improved biogas yield.

### 2.5. Inhibition of Dry AD

One of the principal disadvantages of dry AD is the absence of dilution of inhibitors [12,22]. Six et al. [63] reported that ammonium inhibition was not observed under fermentation of a substrate with a C/N ratio of more than 20. Several authors have pointed out that a C/N ratio between 20 and 30 was necessary to avoid inhibition, and the optimal ratio for the process was 25 [64,65]. High ammonium concentrations are obtained by hydrolyzing compounds containing organic nitrogen. Substrates to which it belongs are complicated for digesting [66]. However, the C/N ratio cannot be used as the main indicator of anaerobic digestion inhibition, because ammonia concentration leads to inhibition and increasing the C/N rate only leads to a decrease in ammonia concentration; the effect of co-fermentation is the same as dilution, i.e., reduction of ammonia content in a substrate.

One of the ways to reduce the effect of ammonium is to remove ammonia from the reactor [3]. However, it has been found that reduction of the inhibitory effect of ammonia was usually reached by reducing the TS concentration using water dilution which leads to an increase in reactor volumes, or by adding chemical reagents which increase the technology cost [67].

Microorganisms that carry out dry AD are more resistant to high ammonium concentrations than those that carry out wet AD. The effect of lower inhibition is supplemented by the worse mixing of inhibitors in the substrate [22]. Thus, Valorga reactors can withstand a load of ammonia up to 3 g/L at a temperature of 40 °C, and Dranco reactors can operate a load of 2.5 g/L at a temperature of 52 °C [22,68]. However, these data are not comparable due to different temperature regimes. Signs of inhibition of volatile fatty acids (VFA) appear at concentrations of acetic acid equal to 5 g/L and butyric acid exceeding 3 g/L [7].

Abbassi-Guendouz determined that anaerobic digestion at extremely high TS concentrations (more than 30%) led to a decrease in the specific surface area and a decrease in methane content [69]. Deublein and Steinhauser [26] argued that under these conditions, growth of microorganisms became suppressed. The influence of an increase in moisture content on the methane fermentation of chicken manure is shown in Figure 3.

### 2.6. The Efficiency of Dry AD

Fruteau de Laclos's research showed that the biogas yield from a single-stage dry ADR that worked on municipal and garden waste was higher than in one that worked in wet conditions [68]. All dry AD technologies provide similar results of biogas production. They vary from 90 mL/g for garden waste to 150 mL/g for food waste [68,70]. This corresponds to a range from 210 to 300 mL/g of TS and about 50 to 70% degradation of TS, which is higher than for wet AD, where it ranges from 40 to 70% [71–73]. However, dry AD technologies have been studied less than wetAD [13].

### 2.7. Technologies of Dry AD

Dry AD can be carried out at an industrial scale when the TS content is 20% or higher, due to the accumulation of solid particles in reactors [22]. More than 90% of the reactors used in Europe are single-stage reactors. Dry and wet AD are distributed equally [70].

Dry AD can be carried out in continuous or batch mode. The companies that offer dry AD technologies that are available in the market, include Dranco, Kompogas, Valorga, and Biocel (see Figure 4). Additionally, installations are divided into stirred and unstirred. These technologies are intended for the digestion of organic components of municipal solid waste, agricultural waste, and food waste.

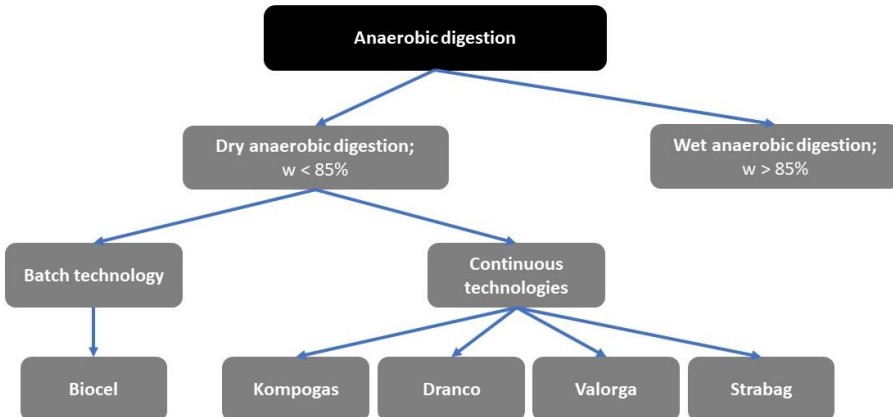

**Figure 4.** Dry AD reactors available in the market.

Dranco reactors are vertical containers that move and mix a substrate from the upper part to the bottom of the reactor [13,22]. They work at a TS concentration equal to 20 to 50% with a hydraulic retention time (HRT) of the reactor in the range of 15 to 30 days [1]. The inoculate is preparing by mixing six parts of substrate with one part of effluent [15,22,74]. The substrate is loaded to the top of the reactor and moves to the conical bottom. After that, the effluent is withdrawn from the reactor, and then solid matter is dehydrated and pressed [13].

The technology works under thermophilic or mesophilic conditions [74,75]. Optimum thermal conditions for fermentation are from 55 to 60 °C. However, in practice, this temperature is not always provided due to inhibition by ammonium [13].

Rapport et al. [13] pointed out that the Dranco technology could provide a biogas yield that equalled 103 to 147 mL/g of wet mass, and Nichols et al. [1] noted that biogas yield ranged from 100 to 200 mL/g. The reactor that provides the Dranco process is shown in Figure 5.

Kompogas reactors are horizontal digesters, which operate continuously with a HRT between 15 and 20 days at a TS substrate content equal from 23 to 28% under thermophilic conditions [13,15]. The Kompogas reactors operate similarly to the Dranco reactors, with the difference that the substrate is mixed by a vortex flow in a horizontal position [13,76].

The horizontal position contributes to the homogenization, degassing, and mixing of particles in the reactor by using impellers [13,22,77]. Such systems require that the TS concentration in the substrate is controlled close to 23%. Particles such as sand and glass accumulate in the reactor when the TS concentration decreases, and it can lead to disruption of mixing when the TS content increases [76]. The biogas output ranges from 110 to 130 mL/g of wet mass [13].

Inhibition by ammonium is characteristic of fermentation of nitrogen-containing substrates. For example, inhibition is typical of systems for protein waste fermentation by Kompogas systems [78]. The reactor that provides the Kompogas process is shown in Figure 6.

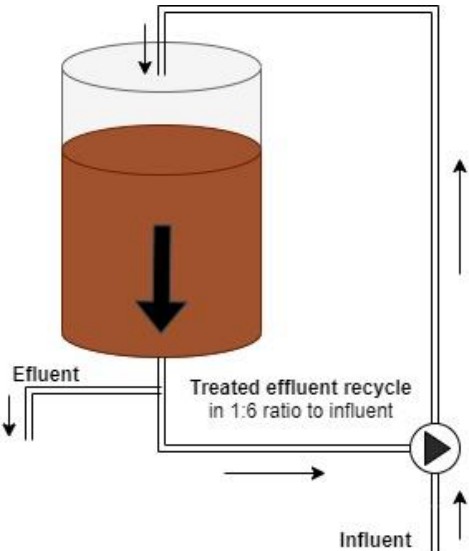

**Figure 5.** The reactor that provides the Dranco process.

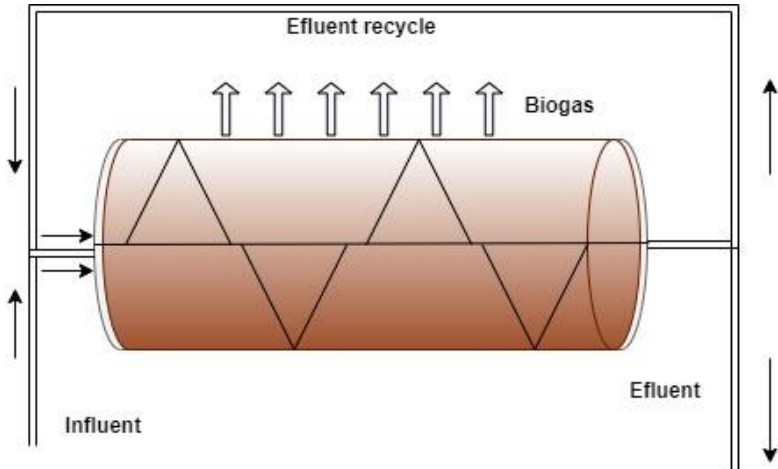

**Figure 6.** The reactor that provides the Kompogas process.

The Strabag technology is very similar to the Kompogas technology but differs by the method of mixing. Mixing is providing by several transversal paddle agitators. This technology is devoted to processing the substrate with a TS content of 15–45% [79]. The reactor that provides the Strabag process is shown Figure 7 [80].

According to Rapport et al., the Valorga technology provides fermentation of the substrate with a TS concentration between 25 and 30% of TS with a HRT up to 3 weeks under thermophilic and mesophilic conditions [13]. Satoto noted that the technology operated at 25 to 32% with HRT between 18 and 25 days, and removed particles with a size of more than 80 mm [15].

The reactor consists of a vertical cylinder with an inner wall extending to about two-thirds of the tank diameter [13,15]. In this case, the substrate moves from one side of the wall to the other side through the partition [15].

Mixing is achieved by injecting the biogas under high pressure into the bottom of the reactor every 15 min through a network of biogas injectors [13,15,68]. Such mixing works satisfactorily, and consequently, there is no need for effluent recycling. The main disadvantage is that the biogas injectors can become clogged, and therefore they require careful maintenance. The Valorga technology is not

suitable for relatively moist waste with a TS concentration lower than 20%, due to the probability of solid particles accumulating in the bottom of the reactor and clogging the injectors [22,81].

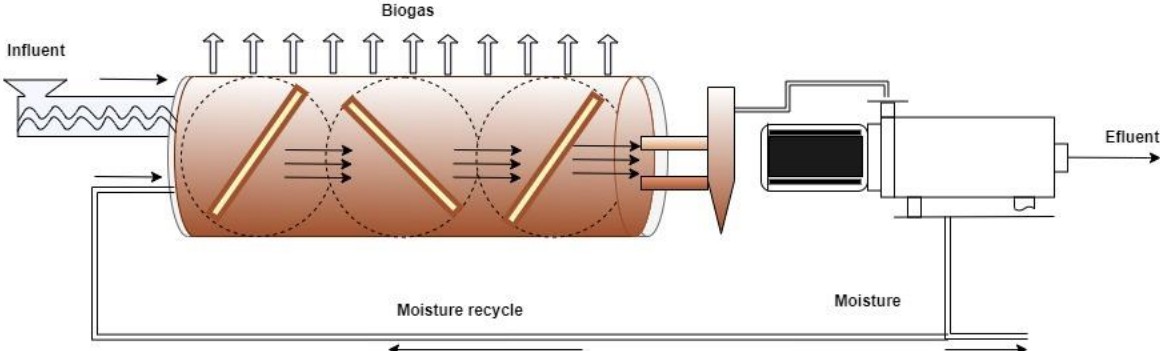

**Figure 7.** The reactor that provides the Strabag process.

Biogas production for Valorga reactors is between 220 and 270 mL/g of municipal waste, which corresponds to 80–160 mL/g of wet substance [1]. The reactor that provides the Strabag process is shown in Figure 8.

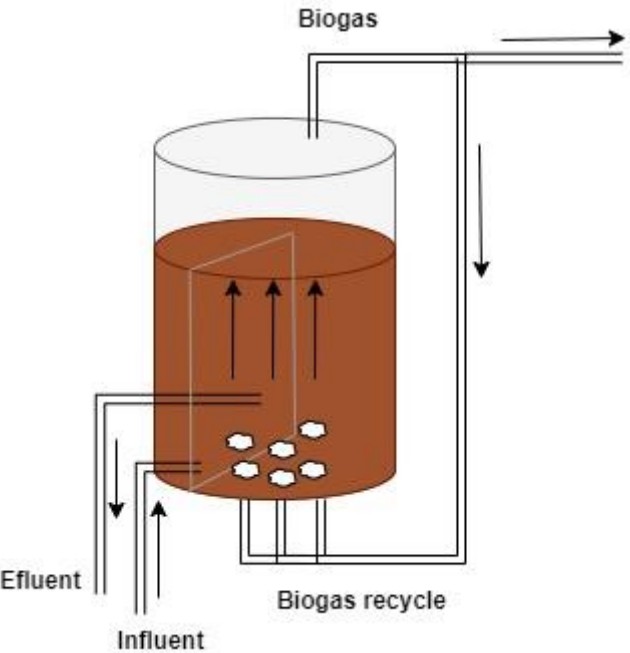

**Figure 8.** The reactor that provides the Strabag process.

One of the other popular types of reactors is the percolation reactors, and the most common type is Biocel [15]. Batch systems are the simplest and cheapest systems. Their main disadvantages are large areas, lower biogas output due to the process of percolation through channel formation, and clogging [22,82]. Biocel plants need ten times more area than continuous systems [82].

Batch systems are loaded once by substrate with a TS concentration of 30 to 40% with or without additions [15,22,83]. According to the Board, the process is stable at 25 to 35% of TS output [13]. The process is carried out in concrete reactors with a perforated floor in mesophilic conditions at a temperature from 35 to 40 °C [13,22,83]. The retention time of Biocel's reactors is between 15 and 21 days [83].

Although Biocel reactors are simple "boxes", they can provide from 50 to 100 times more biogas than landfills [22]. The higher efficiency is related to the fact that percolate is continuously recirculating, which promotes mixing of nutrients, acids, and moisture. In addition, periodic systems are more capable of operating at higher temperatures than those that are characteristic of landfills [15,22].

The main disadvantage of such reactors is the unevenness of gas output and biomass instability. In particular, this is due to the "substrate" and channels clogging, which causes problems with percolation and uneven distribution of moisture. This disadvantage is eliminated by limiting the height of the substrate, i.e., up to 4 m, and by partially mixing the substrate with a filler [13,22,83]. There is also a problem of lack of contact between bacteria and substrate. Such an effect is observed in studies with a TS concentration of more than 35% [12]. Reactors are explosive until opening, and therefore it is necessary to provide proper safety [22].

Investment costs are about 40% lower as compared with continuous-mode technologies [84]. As described earlier, the Biocel technology needs ten times more area than a continuous process because its reactors should be approximately five times smaller, and the reactor loading rate is two times less. Operating costs are similar to other types of fermentation [22]. The reactor that provides the Biocel process is shown in Figure 9.

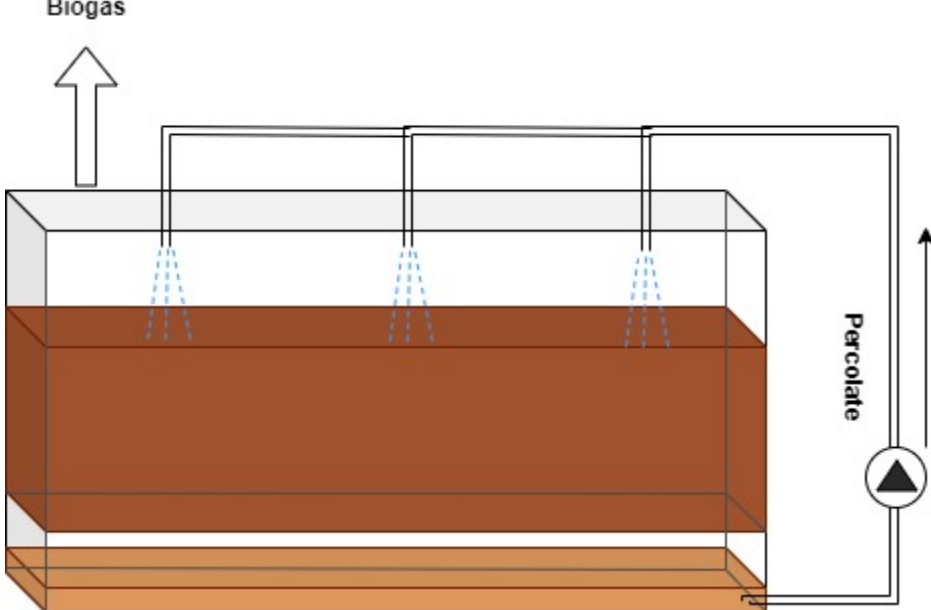

**Figure 9.** The reactor that provides the Biocel process.

Periodic systems are still not widely used but have several advantages, such as easy design and process control, resistance to coarse and solid particles, and low capital costs [85]. Therefore, periodic systems have high potential for implementation in both developing and developed countries, such as the United States [22,82]. A comparison of dry AD technologies is shown in Table 2.

The previously mentioned data and the industrial characteristics in general are all related to organic components of municipal waste or garden waste treatment; however, there are no data on the characteristics of full-scale anaerobic digesters providing dry fermentation of chicken manure. The technologies, potentially, can be used (with or without modification) to provide treatment for high-nitrogen substrates in the case of providing efficiency of technology. Therefore, it seems relevant to analyze the current state of studies devoted to dry anaerobic digestion of chicken manure and summarize its efficiency.

**Table 2.** Comparison of dry AD technologies.

| Technology Type | Biogas Production, mL/g Wet Mass | Temperature Conditions | HRT, Days | TS Content, % | Specific Advantages/ Disadvantages |
|---|---|---|---|---|---|
| | | **Relatively Stirred** | | | |
| **Dranco** | 103–147 (100–200) | Thermophili, mesophilic | 15–30 | 20–50 | No very specific advantages/disadvantages |
| **Kompogas** | 110–130 | Thermophilic | 15–20 | 25–32 | No very specific advantages/disadvantages |
| **STRABAG** | Near to 103.44 | Thermophili, mesophilic | no data | 15–45 | No very specific advantages/disadvantages |
| **Valorga** | 80–160 | Thermophili, mesophilic | 18–25 | 25–32 | -/Clogging of injectors |
| | | **Unstirred** | | | |
| **Biocel** | Twice lower than continuous [86] | Mesophilic | 15–21 | 25–40 | Cheaper, simpler/occupies large areas, the problem of channel formation and clogging, the danger of explosion |

## 3. Dry Anaerobic Digestion of Chicken Manure

A study by Salyuk et al. proved the possibility of dry AD in the range of TS concentration from 16 to 28%. However, the process had a worse performance than wet AD, and there was a tendency to decreased efficiency with TS concentration increase. Moreover, the inhibition under thermophilic conditions was much more intensive than in the mesophilic conditions [16,17,87–89].

Bujoczek et al. [30] reported that adaptation of the anaerobic environment to high ammonia content did not occur at high TS concentrations (19.8% and above). During the dry AD study, methane production was not observing.

Webb et al. [39] investigated batch chicken manure fermentation at a TS concentration of about 16% at a temperature of 30 °C for 45 days. The authors determined that the process inhibited due to VFA and ammonium accumulation.

Farrow [23] conducted a study on chicken manure anaerobic digestion with a TS concentration between 15 and 47% at a temperature of 35 °C. The author reported that there was a biogas yield increase with a VS decrease. A chicken manure VS increase also led to an ammonia concentration increase. Thus, the lowest biogas yield was observed at a TS concentration of 47%. The biogas yield in mesophilic conditions was in the VS range from 100 mL/g at 47% of TS to 140 mL/g at 20% of TS with an ammonia content of 20.3 and 10.2 g/L, respectively. It was inferior at thermophilic temperatures as compared with mesophilic temperatures, due to ammonia.

Abouelenien et al. [90] investigated batch methanogenesis of dry chicken manure with a TS concentration of 22.5% at temperatures of 37, 55, and 65 °C. The authors found that the production of methane occurred only at a temperature of 37 °C. Methane production was about 5 mL/g VS, and ammonium content was about 7 g/L in the first batch [90].

The effect of temperature on chicken manure fermentation at a TS content of 22.25% has been studied. Abouelenien et al. [91] determined that heat did not affect ammonia production. However, a temperature decrease led to a VFA production increase. Thus, the highest VFA content was 84.74 g/L at 35 and 45 °C. There was no effect of temperature on biogas production, but increased temperature led to decreased methane concentration and to increased hydrogen content in biogas. Thus, the methane production was 1.2 mL/g of chicken manure (8.2 mL/g VS) at 35 °C and 0.9 mL/g of chicken manure (6.2 mL/g VS) at 45 °C. The methane concentration in biogas was 10% in both with a pH of 7.2. At higher temperatures, methane production did not occur [91].

Several authors have studied dry AD intensification by conducting control experiments. The data can be used to compare with studies of AD without intensification. Abouelenien et al. [67] studied chicken manure co-fermentation with agricultural wastes at 20% TS. Methane yields in the control

experiment were 136.9 and 129 mL/g of VS at 35 and 55 °C, respectively. The highest ammonium and VFA contents were 3.99 and 17.6 g/L, respectively, under thermophilic conditions [67].

Farrow et al. [23,92] conducted a study on ammonia removal by producing struvite. The biogas yield was 470 mL/g VS in the control experiment at a TS concentration of 20% under mesophilic conditions (at 35 °C). Ammonium concentration was 2.46 g/L at the end of the study.

Methane yield in a control experiment by Markou's study [93] was approximately 117 and 51 mL/g VS with a total nitrogen content of 8 and 10 g/L at a TS content of 15 and 20%, respectively, at 35 °C. The concentration of VFA was 6.5 g/L at a content of TS equal to 15%, and 16 g/L at a TS concentration equal to 20%.

Šinkora et al. [94] researched the process at 38 °C with a substrate TS concentration of about 23%. The authors determined that the optimal loading rate on the reactor was 4.2 g VS/(Lday). For such a loading rate, the average methane yield, over a time period of 32 days, was 246 mL/g VS with a methane concentration of 65.1%. Ammonia concentration was in the range from 1.35 to 2 g/L.

Rajagopal et al. [62] investigated chicken manure fermentation (with straw) at a TS concentration of 30%. Anaerobic digestion with a loading rate of 5.4 g VS/(kg inoculum*day) and 21.6 g VS/(kg inoculum*day) at 20 °C was conducted. The authors reported that methane yield was lower at a higher loading rate. The highest biogas yield was 162 mL/g VS with a methane content of 35% at the loading rate of 5.4 g VS/(kg*inoculum) under down-top mode.

Thus, some results of chicken manure dry AD were successful. Methane yield ranged from 5 to 247 mL/g VS, and the ammonium content ranged from 1.35 to 10.2 g/L in the studies. The highest methane yield was 247 mL/g VS in a study by Šinkora's et al. [94], who reported the lowest ammonium concentration in the range of 1.35–2 g/L at 38 °C. The tendency for decreased methane yield decrease under increased ammonium content was observed. Table 3 presents the results of laboratory studies of chicken manure dry AD.

**Table 3.** Results of laboratory studies of chicken manure dry AD.

| № | TS Content, % | Temperature, °C | Methane Yield, mL/g VS | Ammonium Content, g/L | VFA Content, g/L | Author |
|---|---|---|---|---|---|---|
| | | | **Batch Researches** | | | |
| 1 | 16–28 | 35, 50 | 2–208 | 0.14–9.3 | 0.06–12.6 | Zhadan |
| 2 | 22.5 | 37 | 5 | 7 | no data | Abouelenien |
| 3 | 20 | 35 | 140 | 10,2 | no data | Farrow |
| 4 | 20 | 35 | 217 | 3.5 | no data | Farrow |
| 5 | 20 | 35 | 136.9 | 2.1 | 6.1 | Abouelenien |
| 6 | 20 | 55 | 129 | 3.99 | 17.6 | Abouelenien |
| 7 | 25 | 35 | 8.2 | 16 | 72 ** | Abouelenien |
| 8 | 25 | 45 | 6.2 | 16 | 72 ** | Abouelenien |
| 9 | 25 | 55, 65 | 0 | 16 | 48 ** | Abouelenien |
| 10 | 15 | 35 | 117 | 8 | 6.5 | Markou |
| 11 | 20 | 35 | 51 | 10 | 16 | Markou |
| 12 | 20 | 35 | 470 * | 2.46 | no data | Farrow |
| 13 | 23 | 38 | 247 | 1.35–2 | no data | Šinkora |
| 14 | 30 | 20 | 162 | no data | no data | Rajagopal |
| | | | **Continuous Researches** | | | |
| | | | Not found | | | |

\* biogas yield, mL/g VS. \*\* VFA content, g/kg.

Consequently, the VS concentration and temperature increase led to a decrease of biogas yield and its quality. Chicken manure dry AD was characterized by inhibition, especially in thermophilic conditions, which was associated with a higher proportion of non-dissociated ammonia. However, methane production was possible under dry AD. Methane production from chicken manure under dry AD mesophilic conditions was lower than that obtained in wet AD. Methane production did not occur at thermophilic conditions at all, or it was remarkably weak.

## 4. Ways to Intensify Dry AD of Chicken Manure

### 4.1. Dry Anaerobic co-Fermentation of Chicken Manure

Dry AD can be intensified by ammonia effect reduction, provided by co-fermentation or ammonia removal. To enhance methanogenesis, it is possible to use different co-substrates with high carbon, and low nitrogen content means a high C/N ratio. Attempts to conduct chicken manure co-fermentation with various substrates under dry AD are shown in Figure 10.

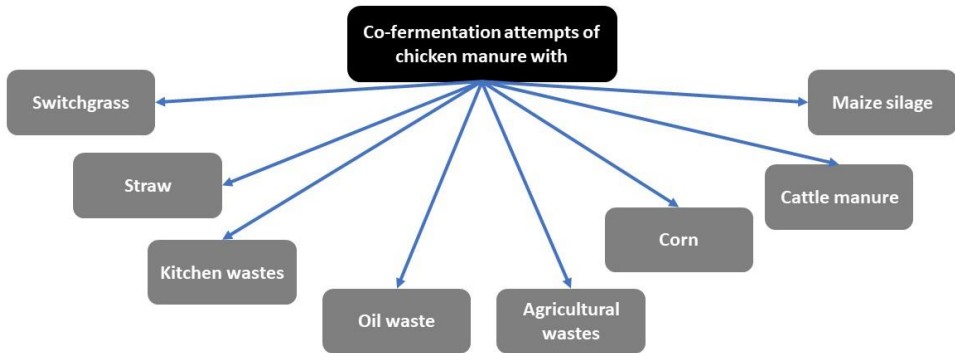

**Figure 10.** Attempts to conduct chicken manure co-fermentation with different substrates under dry AD.

Ahn et al. [95] investigated batch chicken manure co-fermentation with switchgrass (Panicum virgatum) in a ratio of approximately 1:2 with a TS concentration up to 16%, at a temperature of 55 °C. There was inhibition of methane production at ammonium and VFA concentrations of 15 and 9.4 g/L, respectively, at a pH of 5.5. The methane yield was 2 mL/g VS.

Shi et al. [96] conducted chicken manure co-fermentation with straw at a ratio of 2.5:1 and a TS concentration of 20%, in mesophilic conditions at a temperature of 35 °C. The total biogas yield was 4.34 mL/g VS for 15 days. The authors found that the degree of inhibition by ammonium under chicken manure co-fermentation with straw was lower (ammonium content was 0.935 g/L) than that under pig manure co-fermentation with straw (ammonium concentration was 2.729 g/L).

Kukkonen [97] conducted a study on the biological potential of methane production under chicken manure co-fermentation with kitchen waste (actual TS concentration was up to 0.5%) at 35 °C. The author noted that the methane yield under batch mode increased with an increased concentration of kitchen waste. It was possible to achieve a methane production potential of 398.5 mL/g of TS at the ratio of manure to kitchen waste of 1:31.5 (compared to 301 mL/g of TS until chicken manure monofermentation).

Abouelenien [67] conducted a study on chicken manure co-fermentation with agricultural wastes (manioc, coffee, and coconut production wastes) at a ratio of 14:11 and a TS concentration of 20% and at at temperatures of 35 and 55 °C. The authors researched co-fermentation in three stages. The end of each stage was determined by the end of biogas production. After that, the content of the reactor was unloaded by a half and was mixed with a fresh substrate. The results of the first stage were close to the process on an industrial scale [67].

Biogas yields of 406 and 323.4 mL/g of TS were obtained in the mesophilic and thermophilic conditions, respectively, in the first stage. The highest concentration of ammonium and VFA was 2.28 and 0.76 g/L, respectively, under thermophilic conditions.

A lower biogas yield was obtained in the second and third stages due to inhibitor accumulation. Thus, methane production in the third stage did not take place at all, both under thermophilic and mesophilic conditions. The ammonium content under manure co-fermentation was 5.35 g/L and 7.43 g/L, while VFA was 0.43 g/L and 0.47 g/L in thermophilic and mesophilic conditions, respectively [67].

Co-fermentation studies have also been conducted with pretreated chicken manure using the method described by Abouelenien et al. [3]. Methane production occurred in mesophilic conditions

only under pretreated manure co-fermentation at the third step. However, the trend of ammonium accumulation remained. It is worth mentioning that the inhibitor accumulation was more specific for higher temperatures [3].

Callaghan et al. investigated cattle and chicken manure co-fermentation at a ratio of 7:2 at a TS concentration equal to 15% at 35 °C, under batch mode. A methane yield of 70 mL/g of VS was obtained. The ammonium content and the pH at the end of the process were 8.8 g/L and 9.3, respectively. The authors noted that increased chicken manure TS concentration led to a decrease in the process productivity [98].

Jantrania [55] investigated batch anaerobic poultry manure co-fermentation and corn production waste at a TS concentration between 25 and 35%, at 35 °C. A ratio of 2.5:1 of manure to maize waste was used. The author showed the positive effect of increasing the TS concentration on the process. Thus, the maximal methane yield was 42.95 mL/g of VS, with a methane concentration of 59% at a TS concentration equal to 35%. The ammonium concentration was about 20 g/L.

Farrow [23] was able to achieve an increase in biogas output by 22% and 44% while applying chicken manure co-fermentation with silage corn at the ratios of 5:1 and 1.35:1 of grease, respectively, at a TS concentration equal to 20%. The studies were carried out at a temperature of 35 °C. Thus, biogas yield of fermentation with maize silage was 473 mL/g of VS (methane concentration was 246 ml/g VS), and with oil waste, it was 560 mL/g of VS (methane concentration was 270 of mL/g of VS) [23].

A study by Salyuk et al. indicated the negative effect of increased glycerin concentration obtained after biodiesel production, on chicken manure co-fermentation at 35 to 55 °C. The substrate with the ratios of chicken manure to glycerol of 9:1, 8:2, and 7:3 at TS contents equal to 26, 22, and 18%, respectively, were used to provide fermentation. Methane production occurred only when the ratio of chicken manure to glycerol was 9:1 at all moisture content values. Moreover, the results in thermophilic conditions were much worse than in mesophilic conditions. The maximum methane and biogas outputs were observed at a TS concentration of 22%. Thus, the maximum methane output in the mesophilic conditions was 6.34 mL/g of VS with a methane concentration in the biogas of 13.3%, and in thermophilic ones, it was 1 mL/g of VS with 10% of methane. Probably, inhibition was due to substances after biodiesel synthesis [99]. In the works of Foucaul [100], the positive effect of glycerin after biodiesel production was achieved on the compatible fermentation with wastewater from poultry farms.

Patinvoh et al. [101] researched chicken manure co-fermentation with straw at a TS concentration equal to 22.29%, at a C/N ratio of 16.8:1, at 37 °C. They found that the optimum loading rate was 4.2 g of TS/(L*day). Under such a loading rate, the methane yield was 163 mL/g of TS with a methane content of 65.1%. Ammonium concentration was in the range from 1.35 to 2 g/L, and there was no VFA accumulation. The VFA accumulation took place for cases with higher loading rates. However, they found that the limiting factor of manure co-fermentation with straw was the cellulose hydrolysis [102].

Kukkonen [97] provided semi-continuous fermentation of chicken manure and kitchen waste at a ratio of 1:4 with a TS concentration of 26.5%, at a temperature of 35 °C. However, the chicken manure and kitchen waste ratio changed to 1:31.5 with a TS concentration of 18.24%, due to a possible ammonium accumulation. Methane production was stable at a loading rate from 1 to 3 g VS/(L*day). Ammonium and total nitrogen concentrations were 1.3 g/L and 3.3 g/L, respectively, under the loading rate of 3 g VS/(L*day). Methane yield was 388 mL/g VS, which corresponded to 98% of the theoretical value [97].

Chicken manure co-fermentation regularly has better results than chicken manure fermentation as a monosubstrate. The low methane yield was observed only for studies that used switchgrass (2 mL/g VS) [95], straw (4.34 mL/g VS) [96], and glycerol after biodiesel production (6.34 mL/g VS) [99]. In general, methane output in co-fermentation was in the range from 2 to 406 mL/g VS, and ammonia concentration was in the range from 0.935 to 20 g/L. The largest methane yield was 406 mL/g VS, which was demonstrated in the Abouelenien's et al. study [67]. The results of the dry anaerobic co-fermentation of chicken manure with other wastes are presented in Table 4.

**Table 4.** Comparison of the co-fermentation of chicken manure with other wastes.

| № | Co-Substrate | The Ratio of Chicken Manure to the Co-Substrate | TS Content, % | Temperature, °C | Methane Yield, mL/g VS | Methane Content, % | The of Ammonium Content, g/L | The VFA Content, g/L | Intensification of Methane Yield, % | Author |
|---|---|---|---|---|---|---|---|---|---|---|
| | | | | | **Batch Researches** | | | | | |
| 1 | Switchgrass | 1:2 | 16 | 55 | 2 | no data | 15 | 9.4 | no data | Ahn |
| 2 | Straw | 2.5:1 | 20 | 35 | 4.34 | no data | 0.935 | no data | no data | Shi |
| 3 | Corn | 2.5:1 | 35 | 35 | 42.95 | 59 | 20 | no data | no data | Jantrania |
| 4 | Glycerol (biodiesel) | 9:1, 8:2, 7:3 | 26, 22.18% | 35, 55 | 6.34 | 13.3 | - | no data | no data | Shapovalov |
| 5 | Agriculture wastes | 14:11 | 20 | 35 | 406 | no data | 1.39 | 0.47 | no data | Abouelenien |
| 6 | Agriculture wastes | 14:11 | 20 | 55 | 323.4 | no data | 2.28 | 0.76 | 150 | Abouelenien |
| 7 | Cattle manure | 2:7 | 15 | 35 | 70 | no data | 8.8 | no data | 195 | Callaghan |
| 8 | Maize silage | 5:1 / 6.9:1 | 20 | 35 | 246 | 52 | - | no data | no data | Farrow |
| | | | | | **Continuous Researches** | | | | | |
| 9 | Kitchen waste | 1:4 | 26.5 | 35 | 230 | no data | Process significantly inhibited | no data | no data | Kukkonen |
| 10 | Kitchen waste | 1:31.5 | 18.24 | 35 | 388 | no data | 1.3 | no data | no data | Kukkonen |
| 11 | Straw | no data | 22.29 | 37 | 163 | 65.1 | 2 | | no data | Patinvoh |

### 4.2. Dry AD under Ammonium Removal

In addition to co-fermentation, process intensification can be accomplished by reducing the ammonia effect. Ammonia binding, struvite formation, ion-exchange materials (zeolite, gluconate, and iron-contain clay [103–106]) additives, anammox, and denitrification have been used to reduce the ammonia effect [35,107]. However, these removal methods have disadvantages such as additional chemical usage and energy unattractiveness. Several authors have conducted studies to reduce the ammonium nitrogen effects on dry AD. Approaches to reduce the ammonium effect during dry AD are shown in Figure 11.

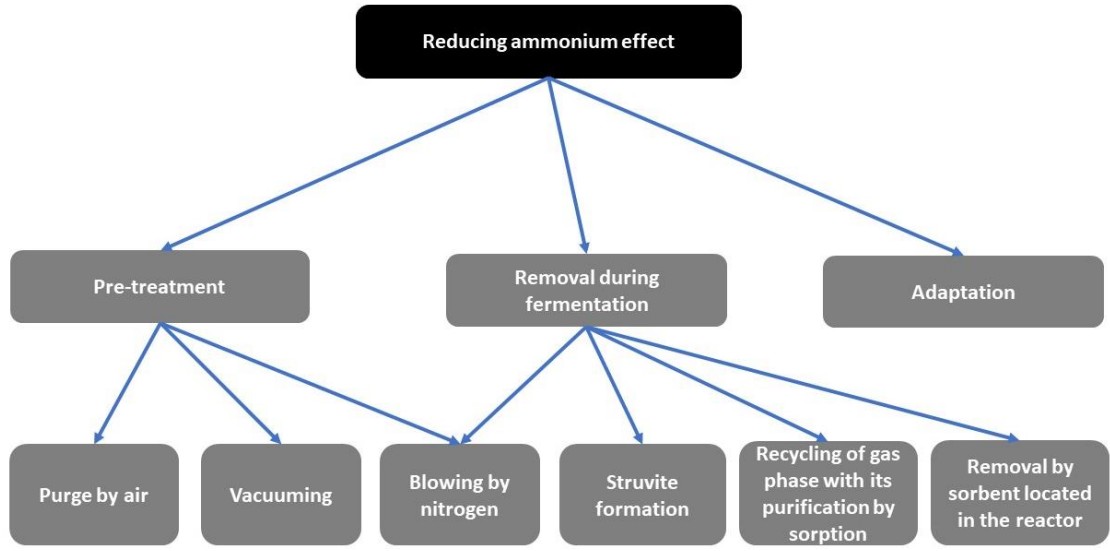

**Figure 11.** Approaches to the effect of ammonium reducing during dry AD.

Markou et al. [93] provided pretreatment in anaerobic conditions for 60 days, at a temperature from 17 to 22 °C, to remove ammonia. After that, the samples were purged by air at 80 °C for 24 h. This method decreased ammonium concentration from 11.2 g/L to 1.86 g/L and total nitrogen from 52.5 g/L to 25.3 g/L. The ammonia nitrogen removal efficiency varied from 62 to 73%, and VFA removal ranged from 41 to 65%. The authors carried out such pretreated manure fermentation at a temperature of 35 °C, at TS concentrations equal to 5, 10, 15, and 20%. Methane yield was 151.3, 153.6, 168.5, and 115.2 mL/g VS at TS concentrations equal to 5, 10, 15, and 20%, respectively.

Farrow et al. [23,92] studied ammonia removal by employing struvite precipitation. The authors carried out research in a test reactor, in a reactor with struvite formation, and in a reactor with the addition of ammonia. Struvite formation and addition of ammonium occurred on the first, seventh, and fourteenth days.

The authors found that the highest biogas yield was 607 mL/g VS under the ammonia removal, which corresponded to an increase in biogas yield of 29%, and the lowest biogas yield was 360 mL/g VS in the reactor with ammonia accumulation, corresponding to a 30% decrease in biogas output. The lowest ammonium content was 0.66 g/L (as compared with 2.46 g/L in test one) in the experiment with struvite formation [23,92].

Another method proposed by Abouelenien et al. [91] was stripping. The approach was based on the stripping of ammonia from the substrate by humidified nitrogen. The process was carried out at a pH of 8–10 of chicken manure, and then brought to a pH of 7 using chloride acid. The authors carried out the removal of ammonia twice after the first fermentation at 65 °C for 8 days, and after the second fermentation at 35 and 55 °C for 75 days. After that, the final fermentation was performed for 55 days under the same conditions. Ammonia removal after the first fermentation reduced the

ammonium nitrogen content from 17 g/L to 2.5 g/L, which corresponded to 85% efficiency. After the second fermentation, the ammonium removal was 74.7% [91].

Methane production did not occur in test reactors under either mesophilic or thermophilic conditions. The amount of produced methane was higher after the second ammonium removal than after the first one. The highest methane yield was 49 mL/g VS at the ratio of 1:2 of chicken manure to inoculum at 35 °C and 103.5 mL/g VS at the ratio of 1:1, at 55 °C [91].

One of the most effective methods of ammonia stripping was proposed by Abouelenien et al. [3]. Ammonia stripping was provided by vacuuming at a temperature of 55 °C and a pH of 8–9. The efficiency of ammonium removal was 82% (the initial ammonium concentration was 16 g/L). However, it was accompanied by moisture loss.

In this regard, the authors offered ammonia stripping using recirculation of the gas phase through the sulfate acid solution during anaerobic digestion. The efficiency of ammonia stripping was 31.4%, and it increased up to 55% when the biogas returned to the bottom with bubbling [3]. However, recycling of biogas through the bottom may be complicated under dry fermentation.

Habibulin studied the reduction of ammonia inhibition effect using the addition of minerals (ground phosphorites). The addition of the minerals in the quantity of 5% was optimal and led to an increase in methane production from 13 to 37 mL at an ammonium concentration of 2.6 g/L [108].

A study by Salyuk et al. proved the effectiveness of ammonia removal using a sorbent (phosphoric acid) located in the reactor and not in contact with the substrate. The main advantages of this technology are positive energy balance and producing additional mono- or diammonium phosphate fertilizer [109–115]. It potentially may be used for dry fermentation.

This method removed ammonium from 3 to 2 g/L, at a TS concentration of 10% and the process was describes as having higher stability. The methane content was 5% higher at the end of the process as compared with the control reactor [109,110].

The proposed methods produced ammonium removal efficiencies from 33 to 80%. The most effective method for ammonia stripping was vacuuming proposed by Abouelenien. However, these methods could decrease economic attractiveness due to additional costs. Ammonia stripping by the sorbent located within the reactor provided a lower ammonia removal rate, but the fermentation was stable without additional energy costs. The efficiency of ammonia removal under dry AD is presented in Table 5.

It is worth noting that there is an alternative way to reduce water consumption during digestion, which is using recirculation of the liquid phase. Several scientific papers have confirmed the possibility of liquid phase recycling [32–34,115–117]. In this case, the removal of inhibitors is needed.

### 4.3. Adaptation during Dry AD

One of the ways to decrease the ammonium effect on anaerobic microorganism is through adaptation of sludge. Adaptation of anaerobic microorganisms to high ammonium concentration has been studied by Abouelenien et al. [90]. The authors carried out fermentation studies using repeated batch cultures at temperatures of 37, 45, 55, and 65 °C. In each batch, half of the reactor capacity was taken out and it was filled with fresh substrate. In general, the authors provided nine batches of fermentation with a total duration of 418 days. The authors found that adaptation to high ammonium concentrations occurred only at 37 °C. The highest biogas and methane yields were 19 mL/g of chicken manure (108 mL/g VS) and 4.4 mL/g of chicken manure (25 mL/g VS), respectively, under the ninth batch of fermentation. There was an ammonium concentration of 14 g/L [90].

**Table 5.** The efficiency of ammonia removal under dry AD.

| Method | The Efficiency of Ammonium Removal, % | Improvement of Efficiency of Methane Yield, % | Author |
|---|---|---|---|
| Purge by air | 62–73 | up to 124 | Markou |
| Struvite formation | 73 | 135 | Farrow |
| Stripping by nitrogen | 74.7 | no data | Abouelenien |
| Vacuuming | 80 | no data | Abouelenien |
| Recirculation of the gas phase | 55 | 40–73 | Abouelenien |
| Addition of minerals | no data | 185 | Habibulin |
| Removal from the gas phase in the reactor by the sorbent | 33 * | 5 * | Salyuk |

* data for wet fermentation.

## 5. Conclusions

The main problem related to anaerobic digestion of chicken manure is the high content of ammonium, which is inhibiting, or even toxic, for microorganisms. This factor defines the lowest level of technology water consumption and, respectively, the quantity of effluent. Dry anaerobic digestion of chicken manure is a prospective way to decrease excess effluent production, however its valorization is characterized by low stability of the process, and therefore it is necessary to optimize the process. Co-fermentation, ammonia removal, and adaptation have been used to decrease the effect of ammonia inhibition. Co-fermentation studies are still characterized by low stability of the process (methane yield ranged from 2 to 406 mL/g VS). The most effective way to decrease ammonium content was vacuuming, which provided 80% efficiency of ammonium content removal, but led to reducing the economic efficiency of the technology. The perspective way to regulate ammonia is through absorption without volatile acid located in the reactor. It decreases ammonia content during wet anaerobic digestion by 33% and has a positive economic effect. Dry anaerobic fermentation of chicken manure is possible, but there is still no efficient way to provide it, and additional research is needed.

**Author Contributions:** Y.S. conceived and designed the analysis, collected the data, contributed data or analysis tools, performed the analysis and wrote the paper; S.Z. conceived and designed the analysis, collected the data, contributed data or analysis tools and performed the analysis; G.B. collected the data, contributed data or analysis tools and performed the analysis; A.S. conceived and designed the analysis, performed the analysis and wrote the paper; V.N. collected the data, performed the analysis and wrote the paper. All authors have read and agreed to the published version of the manuscript.

**Funding:** This research received no external funding.

**Acknowledgments:** We acknowledge editors and reviewers for their valuable suggestions.

**Conflicts of Interest:** The authors declare no conflict of interest.

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
