# Peer review of "Dry Anaerobic Digestion of Chicken Manure: A Review"

_applsci, doi:10.3390/app10217825_

Round 1

Reviewer 1 Report

The article “Dry anaerobic digestion of chicken manure” presented for review concerns the technology of poultry manure processing. At present, much attention is paid to the development of new and increasing the efficiency of the methods used to convert this substrate, mainly in terms of closing the carbon, nitrogen and phosphorus loops in agriculture. The wet anaerobic digestion of chicken manure is more widely researched and described in the literature than dry anaerobic digestion. The authors of the reviewed article conducted an extensive and reliable analysis of technological solutions used for dry anaerobic processing. They emphasized the advantages of this process, but also the limitations this process’ inhibitors. The content of the article is supported by figures and tables that clearly present the analyzed content. The conducted analysis maybe proven useful in the process of planning and implementing research on the development of the dry anaerobic digestion of chicken manure technology.

Author Response

Dear Reviewer, we are thankful to you for taking the time and opportunity to get acquainted with our work! We made edits to the article, which were given by other reviewers, and I am sure that the paper has become better.

With best regards!

Reviewer 2 Report

Dry digestion is an effective method for handling wastes however the process itself suffering from different technical problems. Moreover, handling problematic waste such as chicken manure due to high ammonia content add more problem to the system. In this paper, the authors have studied the different AD technologies and the problems associated with them. Although this work adds to the body of work in this field and has pointed out important aspects of dry digestion regarding the handling of chicken manure, there are different mistakes in the text. Therefore I am asking the authors to address my comments in their response.

detailed comments

  1. In the paper, the problem is not clearly described. It should be explained with clarity why this review is necessary.
  2. The abstract lacking a structure. The abstract should start with a brief introduction and stating the problem. After that, the authors explain what the solution is and how this paper help solving those problems.
  3. The focus of the work is chicken manure however there is no information about chicken manure in the introduction. A brief description of chicken manure with general composition is necessary.
  4. The introduction section is fragmented. Line 37-40 I assume the authors are talking about the wet digestion. It should be clearly stated what process is being discussed.
  5. Instead of “liquid digestion” it is better to use the term “wet digestion”
  6. In the introduction section, the problem should be clearly defined and the authors have to justify why this review is necessary and how it can contribute to resolving the problem.
  7. In section 2.1 authors already discussed the TS content of dry digestion comparing to wet digestion and lower water usage during dry digestion can be deducted directly and citing other articles and comparing different work again is unnecessary in line 70-71. As a matter of the fact, sections 2.1 and 2.2 could be connected better.
  8. Half of section 2.2 is about the drawbacks of dry digestion so the title of section 2.2 is needed to be changed.
  9. Section 2.4 lacking cohesion, it is fragmented; different parts are needed to be reordered. For instance, the paragraph is about the influence of water activity; however, the definition of water activity is mentioned in the third paragraph (line 155) after discussing the effect of it in the first two paragraphs.
  10. Line 211, “This corresponds 210 to 300 ml/g of TS …” please revise the whole sentence and clarify it.
  11. Line 212, use “degradation” instead of “destruction”.
  12. Please correct the nomenclature throughout the text e.g. the unit liter should be written as “L” or “mL”.
  13. In multiple stances, the yield of biogas is mentioned as the volume of gas per weight (mL/g), however, it is important to specify the substrate, and the yield is based on the wet substrate or based on TS or VS of the substrate.
  14. Section 2.7 is one of the main parts of this review however the authors discussed generally only single-stage DAD systems without focusing on chicken manure. In the case of a general comparison of different systems, very detailed reviews are available in literature e.g. https://www.sciencedirect.com/science/article/pii/S1364032114004638. However, It is expected a detailed discussion about DAD systems and their performance for utilizing the chicken manure rather than only a general comparison.
  15. The advantages and disadvantages of the Dranco, Kompogas, STRABAG, and Valorga systems are available in the text but they are missing in table 1. It is important to have them in Table 1.
  16. Section 4.3 is needed to be extended. More references are required for this section. There are different works available in the literature that have reported the effect of adaptation on dry digestion of chicken manure.
  17. In the conclusions section, Point 1 and 2 needed to be revised.

General comment

  1. The consistency of the text needs serious improvement throughout the whole manuscript.
  2. The language of the manuscript needs correction. The text requires proofreading for word usage (incorrect meaning) and sentence construction. I strongly advise the authors to use a language editing service.

Author Response

Dear reviewer, we are very grateful for your time and feedback to improve the quality of this article. That is very important to us, and we tried to take into account your feedback as much as possible. Details are added.

With best regards!

Reviewer 3 Report

The article discussed the design of Dry anaerobic digestion of chicken manure. I found the idea very interesting but the way the manuscript is written below the average. Therefore, the authors consider to revise the following;  

The authors need to conduct extra efforts to enhance the quality of English, the way data is displayed, and the way their results are discussed concerning economic relevance.

The article also has many misleading designs and some picture are not clear.

There are many grammar mistakes that should be corrected. Besides, there are extra spaces between some words. I would advise the authors to get their paper revised extensively by a native English speaker.

I regret to say that the paper in its current format cannot be accepted, especially with weak English language. Here are some comments to help the authors to improve their manuscript. The author failed to explain the molecular details on AD applications in a more comprehensive way.

Improve the resolution of the Fig3, 4,5.

Authors need to include some more information about the role of the anaerobic microbes in dry fermentation or digestion of chicken manure.

In the line number 479, 486, 495 etc check throughout the MS, it should be et al not et all.

Author Response

Dear reviewer, we are very grateful for your time and feedback to improve the quality of this paper. That is very important to us, and we tried to take into account your feedback as much as possible. Details are added.

With best regards!

Reviewer 4 Report

Review and comments to the manuscript ID applsci-968583

Authors: Yevhenii Shapovalov et al.

In my opinion, this manuscript is interesting, and I think that it should be accepted for printing, but not in its current form. Currently, the manuscript requires corrections that are necessary:

1) please remove all shortcomings from work (read it carefully once again)

2) Figs. 3, 4, 10 and 11:  please improve the quality

3) Table 1: What does "-/-" mean? This should be clarified e.g. in the table header

4) Tables 2, 3 and 4:

- What does "-" mean? This should be clarified e.g. in the table header

- We use dots to write numbers in English !!

5) Table 4: What does "*" mean? This should be clarified e.g. in the table header

6) Conclusions: I propose to write conclusions in the form of a summary and a single text.

Author Response

(The authors gave the same response as above.)

Round 2

Reviewer 2 Report

  1. In the previous version of the manuscript, the aim of the article was described in the abstract but in the current version, it is missing. 
  2. line 31 - remove "anaerobic digestion" after the "technology".
  3. line 53 - fix table 1 caption
  4. there are still language mistakes in the text. check the whole text again. e.g line 117- "effect for" - "specially" 
  5. In response 14 it was mentioned "This article is not devoted to discussing the existing technology of dry anaerobic" so 4 pages of the manuscript (about 20%) article are discussing something which is not the aim of this article.  If this is the case please reduce this section, otherwise provide a better discussion connecting these technologies to chicken manure utilization.

Author Response

Dear reviewer, we are very grateful for the helpful recommendations. 

We tried to take them into account as much as possible in order to improve the quality of the article.

With best regards!

Reviewer 3 Report

accept

Author Response

Dear Reviewer,

We are very grateful for your time. Your edits were very valuable and helped to make the article much better.

With best regards!